# SAR:s for the Antiparasitic Plant Metabolite Pulchrol. 1. The Benzyl Alcohol Functionality

**DOI:** 10.3390/molecules25133058

**Published:** 2020-07-04

**Authors:** Paola Terrazas, Efrain Salamanca, Marcelo Dávila, Sophie Manner, Alberto Giménez, Olov Sterner

**Affiliations:** 1Department of Chemistry, Centre for Analysis and Synthesis, Lund University, 22100 Lund, Sweden; paola.terrazas_villarroel@chem.lu.se (P.T.); sophie.manner@chem.lu.se (S.M.); 2Centre of Agroindustrial Technology, San Simón University, 3299 Cochabamba, Bolivia; marcelodavila@fcyt.umss.edu.bo; 3Institute for Pharmacological and Biochemical Sciences, San Andrés University, 3299 La Paz, Bolivia; efrain_salamanca@hotmail.com (E.S.); agimenez@megalink.com (A.G.)

**Keywords:** *Trypanozoma cruzi*, *Leishmania amazoniensis*, *Leishmania brasiliensis*, pulchrol, benzo[*c*]chromenes, SAR

## Abstract

Pulchrol (**1**) is a natural benzochromene isolated from the roots of *Bourreria pulchra*, shown to possess potent antiparasitic activity towards both *Leishmania* and *Trypanozoma* species. As it is not understood which molecular features of **1** are important for the antiparasitic activity, several analogues were synthesized and assayed. The ultimate goal is to understand the structure–activity relationships (SAR:s) and create a QSAR model that can be used for the development of clinically useful antiparasitic agents. In this study, we have synthesized 25 2-methoxy-6,6-dimethyl-6*H*-benzo[*c*]chromen analogues of **1** and its co-metabolite pulchral (**5a**), by semi-synthetic procedures starting from the natural product pulchrol (**1**) itself. All 27 compounds, including the two natural products **1** and **5a**, were subsequently assayed in vitro for antiparasitic activity against *Trypanozoma cruzi*, *Leishmania brasiliensis* and *Leishmania amazoniensis*. In addition, the cytotoxicity in RAW cells was assayed, and a selectivity index (SI) for each compound and each parasite was calculated. Several compounds are more potent or equi-potent compared with the positive controls Benznidazole (*Trypanozoma*) and Miltefosine (*Leishmania*). The compounds with the highest potencies as well as SI-values are esters of **1** with various carboxylic acids.

## 1. Introduction

Parasites belonging to the *Trypanozoma* and *Leishmania* genera affect a large proportion of the world’s inhabitants, approximately 12 and 8 million people, respectively, especially in the tropic regions. Surprisingly little is known about these organisms, and there are few efficient drugs against them on the market. This is partly due to the fact that the affected regions in general are poor and cannot pay for advanced medications, and these parasitic diseases are consequently labelled as neglected [1,2]. Benzochromenes are polycyclic aromatic compounds formed by the fusion of a benzene ring with a chromene moiety [3]. The benzochromene structure is found in many natural products isolated from plants [4,5,6,7,8], lichens and fungi [9], and a wide range of biological activities such as antibacterial [5,10], antioxidant and anticancer activities have been reported for natural benzochromenes [9,11]. They have been reported to be able to intercalate with DNA [11,12], and to bind selectively to the estrogen receptor ERβ [13]. Several cannabinoids are benzochromenes and bind to the cannabinoid receptors CB1 and CB2 [5,14].

This work focuses on the natural benzo[*c*]chromene pulchrol (**1**), isolated from the heptane fraction of the roots of *Bourreria pulchra* together with the corresponding aldehyde pulchral (**5a**) [4]. *B. pulchra*, also known locally as “bakalche” and “azar del monte”, is part of the traditional medicine of the Yucatan peninsula (Mexico) and has been used to treat cutaneous diseases, injuries, viral infections and fevers [15,16]. We have reported antiparasitic activity against *Leishmania* (*L. brasiliensis*, *L. amazoniensis*, *L. Mexicana*) and *Trypanozoma* (*T. cruzi*) species of **1** and **5a** (see Figure 1 for structures) in vitro [4], and more recently an ethanol extract of *B. pulchra* was reported to possess potent activity against *T. cruzi* [17]. As only limited amounts of the compounds **1** and **5a** are available from natural sources, which is insufficient for a complete biological characterization, a synthetic route to both compounds was developed. To facilitate the semi-synthesis of new analogues starting from pulchrol (**1**), the synthetic procedure of **1** was further developed in order to improve the yields and increase the throughput. In this study, we present the synthesis of 25 pulchrol analogues systematically varied in the benzyl alcohol region while keeping the 2-methoxy-6,6-dimethyl-6*H*-benzo[*c*]chromen structure intact, and report their in vitro antiparasitic activity against *T. cruzi* epimastigotes and *L. amazoniensis*, as well as *L. brasiliensis* promastigotes. In addition, their cytotoxicity in a murine macrophage RAW cell line was investigated, in order to get a general overview of the selectivity of the assayed compounds. The SAR:s that this and future studies suggest will be used for the development of a QSAR model to enable the design of more potent and selective antiparasitic drug candidates. In addition, improved understanding of the molecular targets of the parasites could be obtained, facilitating the developing novel antiparasitic agents.

## 2. Results and Discussion

### 2.1. Improvements of the Synthetic Route to Pulchrol *(**1**)*

Several synthetic strategies to prepare benzo[*c*]chromenes have been reported in the literature [18,19]. However, as the focus in this study is on the benzyl alcohol functionality of **1**, we essentially relied on the synthetic manipulation of pulchrol (**1**) itself. An important intermediate in the synthesis of **1** is the biaryl intermediate **2** (see Scheme 1), which can be transformed into **1** by intramolecular cyclization. **2** can be obtained by a metal catalyzed Suzuki-Miyaura cross coupling reaction, using phenyl boronic acid and a *o*-halo-benzoic acid [3,13,14,20,21]. Other synthetic routes to **2** are by a dicarbonyl cycloaddition to chromenes [20], or by the direct intramolecular biaryl formation from phenylbenzyl ethers, as the subsequent cyclization can be performed by metal catalysis [20,22,23] or by a radical reaction promoted by K*t*BuO [20,24]. A synthetic route to pulchrol was reported in 2014, in which the biaryl formation was achieved with a Suzuki-Miyaura coupling and the final cyclization was acid catalyzed [18]. The procedure for preparing **1** in this study was based on this strategy, although we introduced conditions that are milder, increased the yields, and shortened the reaction times.

As in the reported synthesis [18], commercially available 3-iodo-4-(methoxycarbonyl)benzoic acid (also known as 1-methyl-2-iodoterephthalate) was used as the starting material. This was reduced to the corresponding benzyl alcohol (methyl 4-(hydroxymethyl)-2-iodobenzoate) using a borane-tetrahydrofuran complex in THF as solvent, at 0 °C (step a). The benzylic hydroxyl group was protected with *tert*-butyldiphenylchlorosilane (TBDPSCl) using pyridine as solvent (step b), and this intermediate was then coupled with 2,5-dimethoxyphenyl boronic acid using palladium-tetrakis(triphenylphosphine) (Pd(PPh_3_)_4_) as catalyst and K_2_CO_3_ as base, in dimethoxyethane (DME)/H_2_O 4:1, to yield the biaryl precursor of **2** (step c). This coupling was reported to work successfully at 120 °C for a period of 14 h in 83% yield [18]. However, microwave-assisted Suzuki couplings have been shown to reduce the reaction time and increase the yields [25], and we obtained 85–90% yield at 100 °C in 30 min using a microwave reactor. Instead of MeMgBr in tetrahydrofuran (THF) at 40 °C for 18 h, the methyl ester group was transformed to the tertiary alcohol **2** by two equivalents of methyllithium (MeLi) in THF at 0 °C for 8 h in similar yields, and the milder conditions produced a cleaner product that was considerably easier to purify (step d). By using a larger excess of hydroiodic acid (HI) (10 equiv) for the cyclization of **2** to **1** (step e) we completely avoided the formation of cannabidiol-type biaryl by-products and obtained the deprotected product directly.

### 2.2. Transformations of the Benzylic Alcohol Functionality

Figure 2 summarizes the structure types of the analogues prepared from **1**. See the experimental part for details about how each individual analogue was prepared. The biological activities are given in Table 1.

Table 2 and Table 3 give the 1D ^1^H and ^13^C-NMR shifts of the assayed compounds. In general, 1-H is a doublet (d) with coupling constant (*J*) close to 2 Hz, 3-H is a doublet of doublet (dd) with *J* = 9 and 2 Hz, while 4-H is a d with *J* close to 9 Hz. 10-H is a d with *J* close to 2 Hz, 8-H is a dd with *J* = 8 and 2 Hz, while 7-H is a d with *J* close to 8 Hz.

### 2.3. Antiparasitic Activity of Pulchrol (1), the Starting Point

Pulchrol (**1**) and pulchral (**5a**) have previously been shown to possess antiparasitic activity, **1** towards *T.cruzi* epimastigotes and three strains of *Leishmania* promastigotes (*L. mexicana*, *L. brasiliensis* and *L. amazoniensis*), and **5a** towards *L. brasiliensis* and *L. amazoniensis* [4]. The IC_50_ value of **1** against *T.cruzi* in this investigation was 18.5 μM (see Table 3), and it is thereby equipotent with the positive control benznidazole (19.2 μM). Benznidazole is currently on the market for the treatment of Chagas disease, caused by *T. cruzi*, under the trade names Rochagan and Radanil. The potency of pulchrol against the *Leishmania* parasites, with the IC_50_ values 59.2 μM for *L. brasiliensis* and 77.7 μM for *L. amazoniensis*, is more moderate, although promastigotes of *L. mexicana* (not part of this investigation) are more sensitive with an IC_50_ value of 17 μM [4]. As the chemical structure of pulchrol (**1**) does not raise any red flags, it contains no functionalities that are associated with reactivity or unspecific biological activity, we were motivated to synthesize and assay analogues of **1**. This study focuses on the importance of the benzyl alcohol functionality for the biological activity, and the natural products (**1** and **5a**) together with 25 analogues were prepared as discussed above, and assayed. The assays against *T. cruzi* epimastigotes and *L. amazoniensis* as well as *L. brasiliensis* promastigotes essentially follows the protocol used in previous investigations, but the cytotoxicity in a mammalian murine macrophage RAW cell line was also assayed in order to get an impression of the compounds selectivity for the parasites over a mammalian cell line. The biological results are presented in Table 1.

### 2.4. Antiparasitic Activities towards Trypanozoma cruzi Epimastigotes

To determine the importance of the hydroxyl group for pulchrol’s activity, the 9-methyl analogue (**3a**) was prepared and found to be considerably less active (IC_50_ = 51.1 μM) compared to **1**. With the intention of mimicking the Van der Waals interactions around the benzylic carbon, the hydroxyl group was replaced by a chlorine (**3b**), but this analogue was also less potent than **1** (IC_50_ = 38.1 μM). It would appear to be beneficial to have an oxygen in the benzylic position, although **3a** and **3b** are by no means inactive. To evaluate if the hydroxyl group acts as a hydrogen bond donor, the methyl ether **3c** was prepared and assayed. It is slightly less potent compared to **1** (IC_50_ = 24.6 μM), indicating that the hydroxyl group is more a hydrogen bond acceptor than donor. However, the two bulkier ethers **3d** and **3e** were actually more potent than **1** (IC_50_ = 12.9 and 9.0 μM, respectively), suggesting that there also is a lipophilic pocket close to the binding site of the benzylic moiety in a target protein. In addition, **3d** and **3e** show an improved SI compared to **1** and **3c**. Somewhat surprisingly, the isopropylamino analogue **3f** is considerably less potent and selective compared to the isopropyl ether **3d**, while the isobutyl and isopentyl analogues **3g** and **3h** (IC_50_ = 15.4 and 5.9 μM, respectively) are as potent as the bulkier ethers but less selective.

Moving to the pulchrol esters **4a**–**4l**, it is clear that this group of analogues is interesting as most of them are more potent and selective compared to **1**. For the saturated esters **4a**–**4i** it is especially those with branched alkyl groups that are good, with the 3-methylbutanoic acid ester **4e** standing out with IC_50_ = 4.2 μM and SI = 6.7. All the unsaturated esters **4j**–**4l** prepared and assayed are potent and selective, indicating that a π-π interaction with the binding pocket is favourable. The furan-2-carboxylic acid ester **4k** is actually the most potent (IC_50_ = 3.8 μM) and the most selective (SI = 7.9) towards *T. cruzi* of all analogues prepared in this investigation, and has considerably better antiparasitic activity towards *T. cruzi* epimastigotes compared to the positive control Benznidazol (see Table 1). Also the selectivity is noteworthy, as it is twice that of the positive control.

Among the 1’-carbonyl analogues (**5a**–**5e**) included in this study, the aldehyde **5a** and the methyl ketone **5b** can be compared to **1**, both with respect to potency and selectivity. However, the carboxylic acid **5c**, the methyl ester **5d** and especially the amide **5e** are less potent, although the carboxylic acid **5c** is considerably less cytotoxic than the others. The *N*-hydroxy-9-carboximidamide **6** was obtained as a by-product and assayed; it did not show any interesting activities, although it was more potent than the 9-carboxamide **5e**. If anything, analogue **6** underlines the importance of a lipophilic component at the benzylic moiety.

### 2.5. Antiparasitic Activities towards Leishmania brasiliensis Promastigotes

As for *T.cruzi*, transforming the benzylic alcohol moiety to a methyl group is not beneficial, and **3a** was found to be slightly less potent than **1**. On the contrary, the benzyl chloride (**3b**) showed both an interesting potency (IC_50_ = 17.1 μM) and selectivity (SI = 3.5), suggesting that the presence of an oxygen in the benzylic position is less important. The differences in antiparasitic effects of **3b** in *T. cruzi* epimastigotes and *L. brasiliensis* promastigotes indicate that the molecular targets in the two species are different. Although the two ethers **3c** and **3d** are slightly more potent than **1**, the 4-methylpentyl ether **3e** is considerably less potent and the positive effect of bulky ethers observed for *T. cruzi* is not seen with *L. brasiliensis*. However, for the secondary amines **3f**–**3h** the trend is identical, and the isopentylamino analogue **3h** (IC_50_ = 15.9 μM) is one of the most potent against *L. brasiliensis*. Most of the esters **4a**–**4l**, except for **4h**, are more potent compared to **1**, and among the saturated esters there is again a tendency that branched alkyl groups are better than straight. The aromatic esters **4k** and **4l** are potent towards *L. brasiliensis* as well, **4k** (IC_50_ = 12.8 μM) is as potent as the positive control, although the selectivity observed towards *T. cruzi* is less prominent. The vinyl ester **4j** is the most potent and selective towards *L. brasiliensis*, with IC_50_ = 5.7 μM and SI = 7.0, overshadowing the positive control. For the 1’-carbonyl analogues, the aldehyde **5a**, the methyl ketone **5b** and especially the methyl ester **5d** are more potent, while the carboxylic acid **5c** and the *N*-hydroxy-9-carboximidamide (**6**) are comparable to **1**. The carboxamide **5e** is considerably less potent than **1**.

### 2.6. Antiparasitic activities towards Leishmania amazoniensis promastigotes

For *L. amazoniensis*, too, the replacement of the benzylic alcohol moiety for a methyl group (**3a**) does not improve the antiparasitic activity, and as for *L. brasiliensis*, a chlorine substituent in this position (**3b**) increases the potency more than two-fold. Ethers of pulchrol (**1**) are more potent; the methyl and isopropyl ethers (**3c** and **3d**) only slightly, but the 4-methylpentyl ether **3e** more clearly. This is in contrast to the poor potency of **3e** towards *L. brasiliensis*. For the secondary amines the sensitivity of *L. amazoniensis* follows that observed already for T. cruzi and *L. brasiliensis*, lower potency for the isopropylamino analogue **3f** and higher for the isobutyl- and isopentylamino analogues **3g** and **3h**. For the esters **4a**–**4l**, the results follow those obtained with *L. brasiliensis* (*vide supra*) closely. There is only one exception, the 3-cyclopentylpropanic acid ester **4h**, which displays a more expected potency towards *L. amazoniensis* than towards *L. brasiliensis* (*vide supra*). Again, the vinyl ester **4j** is the most potent towards *L. amazoniensis* among the esters, and actually among all compounds assayed here, and as the SI value is 5.8 it is also by far the most selective compound. Among the 9-carbonyl analogues, the aldehyde **5a** and the methyl ketone **5b** are more potent, while the methyl ester **5d** and the *N*-hydroxy-9-carboximidamide (**6**) are only slightly more potent. The carboxylic acid **5c** and the carboxamide **5e** are both considerably less potent compared to **1**.

## 3. Materials and Methods

### 3.1. General

^1^H-NMR spectra (400 MHz) and ^13^C-NMR spectra (100 MHz) were recorded with a Bruker Avance II (Bruker Biospin AG, Industriestrasse 26, 8117 Fällanden, Switzerland) in CDCl_3_. The individual 1D signals were assigned using 2D NMR experiments (COSY, HSQC, HMBC). The chemical shifts are given in ppm with the solvent signal as reference (7.27 ppm for ^1^H and 77.0 for ^13^C). Infrared spectra were recorded with a Bruker Alpha-P FT/IR instrument (Bruker Biospin AG, Industriestrasse 26, 8117 Fällanden, Switzerland) with a Diamond ATR sensor as films, and the intensities are given as vw (very weak), w (weak), m (medium), s (strong) and vs (very strong). High-resolution mass spectra (HRMS) were recorded with a Waters XEVO-G2 QTOF equipment (Waters Corp, Milford, Worcester County, Massachusetts, United States), with electrospray ionization (ESI). Synthetic reactions were monitored by TLC using alumina plates coated with silica gel and visualized using either UV light and/or spraying/heating with vanillin/H_2_SO_4_. Flash chromatography was performed with silica gel (35–70 μm, 60 Å). THF was distilled from sodium, acetonitrile was distilled from CaH_2_ and other reaction solvents were dried with Al_2_O_3_. Commercially available compounds were obtained from Aldrich.

### 3.2. Synthetic Procedures

*Methyl 4-(hydroxymethyl)-2-iodobenzoate* (intermediate in the synthesis of **1**) BH_3_-THF (1 M, 65 mL, 65.0 mmol) was slowly added to a stirred solution of 3-iodo-4-(methoxycarbonyl)benzoic acid (5 g, 16.3 mmol) in dry THF (200 mL) at 0 °C. After 30 h, saturated aqueous NaHCO_3_/H_2_O was added, the aqueous phase was extracted with ethyl acetate (3 × 200 mL) before drying (Na_2_SO_4_) and removal of solvent under reduced pressure. Purification by column chromatography (SiO_2_, 4:6 heptane/ethyl acetate) gave (3.74 g, 78%) of the pure product as yellow crystals, identical to that previously reported [18].

*Methyl 4-(((tert-butyldiphenylsilyl)oxy)methyl)-2-iodobenzoate* (intermediate in the synthesis of **1**), TBDPSCl (4.0 mL, 15.3 mmol) was added to a stirred solution of methyl 4-(hydroxymethyl)-2-iodobenzoate (prepared as described above, 3.74 g, 12.8 mmol) in pyridine (60 mL) at rt. After 24 h, saturated aqueous NH_4_Cl/H_2_O was added and the aqueous phase was extracted with diethyl ether (3 × 200 mL), the organic phase was washed with brine (2 × 500 mL) before drying (Na_2_SO_4_) and removal of solvent under reduced pressure. Purification by column chromatography (SiO_2_, 20:2 heptane/ethyl acetate) gave (6.1 g, 89%) of the product as white crystals, identical to that previously reported [18].

*Methyl 5-(((tert-butyldiphenylsilyl)oxy)methyl)-2’,5’-dimethoxy-[1,1’-biphenyl]-2-carboxylate* (intermediate in the synthesis of **1**), 2,5-dimethoxyphenylboronic acid (155 mg, 0.85 mmol), K_2_CO_3_ (394 mg, 2.85 mmol) and tetrakis(triphenylphosphine)palladium(0) (115 mg, 0.1 mmol), were added to a stirred solution of methyl 4-(((tert-butyldiphenylsilyl)oxy)methyl)-2-iodobenzoate (prepared as described above, 300 mg, 0.57 mmol) dissolved in 4:1 DME/water (15 mL). The mixture (contained in a microtube) was degasified under vacuum/N_2_ at −78 °C five times. The microwave reaction conditions were 100 °C, high pressure, and 10 s of pre-stirring. After 30 min in the microwave reactor, the mixture was filtered through a plug of celite and washed with ethyl acetate (250 mL) before drying (Na_2_SO_4_) and removal of solvent under reduced pressure. Purification by column chromatography (SiO_2_, 20:3 heptane/ethyl acetate) gave the pure product as a yellowish wax (263 mg, 86%), identical to that previously reported [18].

*2-(5-(((tert-butyldiphenylsilyl)oxy)methyl)-2’,5’-dimethoxy-[1,1’-biphenyl]-2-yl)propan-2-ol* (**2**), Methyl lithium (3 M, 3 mL, 8.8 mmol) was added to a stirred solution of methyl 5-(((*tert*-butyldiphenylsilyl)oxy)methyl)-2’,5’-dimethoxy-[1,1’-biphenyl]-2-carboxylate (prepared as described above, 1.2 g, 2.2 mmol) in dry THF (70 mL) at 0 °C. After 8 h, saturated aqueous NH_4_Cl/H_2_O was added, and the aqueous phase was extracted with ethyl acetate (3 × 100 mL). The organic phase was dried (Na_2_SO_4_) and after the removal of the solvent under reduced pressure, column chromatography (SiO_2_, 20:4 heptane/ethyl acetate 20:4) gave the pure product as yellowish wax (0.88 g, 74%), identical to that previously reported [18].

*Pulchrol, or (2-methoxy-6,6-dimethyl-6H-benzo[c]chromen-9-yl)methanol* (**1**), HI (55% wt, 0.75 mL, 5.5 mmol) was added to a stirred solution of **2** (50 mg, 0.09 mmol) in MeCN (12.5 mL) at rt. After 70 min, saturated aqueous Na_2_S_2_O_3_/H_2_O was added, and the aqueous phase was extracted with ethyl acetate (3 × 12.5 mL), the organic phase was dried (Na_2_SO_4_) and after the removal of the solvent under reduced pressure the crude product was purified by column chromatography (SiO_2_, 20:4 heptane/ethyl acetate) to give pure **1** as a yellowish wax (16.3 mg, 65%). NMR data are shown in Table 2 and Table 3. HRMS-ESI+ (*m/z*): [M + H]^+^ calcd for C_17_H_19_O_3_, 271.1334; found, 271.1339. IR (cm^−1^): 3402 (br, OH), 2933 (w, CH), 1503 (s), 1463 (m), 1281 (w), 1217 (vs), 1156 (w), 1040 (m), 820 (w).

*2-Methoxy-6,6,9-trimethyl-6H-benzo[c]chromene* (**3a**), Et3SiH (0.16 mL, 1.0 mmol) and PdCl2 (73.5 mg, 0.4 mmol) were added to a stirred solution of 1 (56 mg, 0.2 mmol) in EtOH (25 mL) at rt. After 150 min, ethyl acetate was added and the mixture was filtered through a plug of celite and washed with ethyl acetate (500 mL). After drying of the organic phase (Na_2_SO_4_) and the removal of solvent under reduced pressure, purification by column chromatography (SiO_2_, 20:1 heptane/ethyl acetate) gave pure 3a as a colorless oil (48 mg, 92%). 1H and 13C-NMR data are shown in Table 2 and Table 3. HRMS-ESI+ (*m*/*z*): [M + H]^+^ calcd for C_17_H_19_O_2_, 255.1385; found, 255.1392. IR (cm^−1^): 2931 (m, CH), 1723 (vw), 1504 (m), 1217 (s), 1129 (m), 755 (vs).

*9-(chloromethyl)-2-methoxy-6,6-dimethyl-6H-benzo[c]chromene* (**3b**), *p*-Toluensulfonylchloride (20.9 mg, 0.11 mmol), DMAP (10.7 mg, 0.09 mmol) and pyridine (8.9 uL, 0.11 mmol) were added to a stirred solution of **1** (19.8 mg, 0.07 mmol) in CH_2_Cl_2_ (10 mL) at 0 °C. After 24 h, diethyl ether was added, the mixture was stirred for 30 min, and filtered. The filtrate was washed once with saturated aqueous NaHCO_3_/H_2_O and once with brine, before drying (Na_2_SO_4_) and removal of the solvent under reduced pressure. The purification of the crude product by column chromatography (SiO_2_, 20:2 heptane/ethyl acetate) gave **3b** as a colorless oil (5 mg, 23%). ^1^H and ^13^C-NMR data are shown in Table 2 and Table 3. HRMS-ESI+ (*m/z*): [M + NH_4_]^+^ calcd for C_17_H_18_O_2_Cl, 289.0995; found, 289.0992. IR (cm^−1^): 2938 (w, CH), 1723 (w), 1504 (s), 1424 (m), 1281 (m), 1218 (vs), 1157 (m), 1114 (w), 1040 (m), 814 (w).

*General procedure to prepare compounds***3c**–**3e**, CuBr_2_ (0.1 equiv) was added to 1 (1 equiv) in the corresponding alcohol as solvent (1 mL), at *re*flux. After 12 h, the mixture was cooled to rt, whereafter 10% Na_2_CO_3_/H_2_O was added and the aqueous phase was extracted with CHCl_3_ (3 × 1 mL). The mixture was filtered through a pad of silica gel and washed with heptane before the solvent was removed from the organic phase under reduced pressure. Purification of the crude products by column chromatography (SiO_2_, 40:1 heptane/ethyl acetate) gave the desired products **3c**–**3e**.

*2-Methoxy-9-(methoxymethyl)-6,6-dimethyl-6H-benzo[c]chromene* (**3c**), Yield 33%. ^1^H and ^13^C shifts are shown in Table 2 and Table 3. HRMS-ESI+ (*m/z*): [M + H]^+^ calcd for C_18_H_21_O_3_, 285.1491; found, 285.1494. IR (cm^−1^): 2941 (w, CH), 1502 (s), 1424 (m), 1262 (vs), 1105 (w), 1040 (m), 817 (m).

*9-(Isopropoxymethyl)-2-methoxy-6,6- dimethyl-6H-benzo[c]chromene* (**3d**), Yield 26%. ^1^H and ^13^C shifts are shown in Table 2 and Table 3. HRMS-ESI+ (*m/z*): [M + H]^+^ calcd for C_20_H_24_O_3_Na, 335.1623; found, 335.1627. IR (cm^−1^): 2931 (w, CH), 1726 (w), 1503 (m), 1463 (w), 1428 (w), 1280 (m), 1218 (s), 1111 (s), 1040 (w), 822 (vw), 756 (vs), 704 (w).

*2-Methoxy-6,6-dimethyl-9-(((4-methoxypentyl)oxy)methyl)-6H-benzo[c]chromene* (**3e**), Yield 41%. ^1^H and ^13^C shifts are shown in Table 2 and Table 3. HRMS-ESI+ (*m/z*): [M + H]^+^ calcd for C_23_H_30_O_3_Na, 377.2093; found, 377.2100. IR (cm^−1^): 2953 (m, CH), 1503 (s), 1465 (w), 1424 (w), 1281 (w), 1218 (vs), 1156 (w), 1098 (m), 1042 (m), 818 (w).

*General procedure to prepare***3f**–**3h**, [Cp*IrCl_2_]_2_ (0.005 eq), NaHCO_3_ (0.005 eq), and the corresponding amine (2 eq) were added to a stirred solution of **1** (1 eq) in dry toluene (2 mL), at 100 °C. After 36 h, the solvent was removed under reduced pressure, and purification of the crude product by column chromatography (SiO2, 20:3:0.1 heptane/ethyl acetate/triethylamine) gave the desired products **3f**–**3h**.

*N-((2-Methoxy-6,6-dimethyl-6H-benzo[c]chromen-9-yl)methyl)propan-2-amine* (**3f**), Yield 25%. ^1^H and ^13^C shifts are shown in Table 2 and Table 3. HRMS-ESI+ (*m/z*): [M + H]^+^ calcd for C_20_H_26_NO_2_, 312.1963; found, 312.1964. IR (cm^−1^): 2929 (w, CH), 1504 (w), 1218 (m), 1155 (w), 1041 (w), 755 (vs).

*N-((2-Methoxy-6,6-dimethyl-6H-benzo[c]chromen-9-yl)methyl)-2-methylpropan-1-amine* (**3g**), Yield 27%. ^1^H and ^13^C shifts are shown in Table 2 and Table 3. HRMS-ESI+ (*m/z*): [M + H]^+^ calcd for C_21_H_28_NO_2_, 326.2120; found, 326.2120. IR (cm^−1^): 2954 (w, CH), 1503 (w), 1217 (m), 1156 (w), 1041 (w), 755 (vs).

*N-((2-Methoxy-6,6-dimethyl-6H-benzo[c]chromen-9-yl)methyl)-3-methylbutan-1-amine* (**3h**), Yield 33%. ^1^H and ^13^C shifts are shown in Table 2 and Table 3. HRMS-ESI+ (*m/z*): [M + H]^+^ calcd for C_22_H_30_NO_2_, 340.2277; found, 340.2275. IR (cm^−1^): 2957 (m, CH), 1503 (s), 1422 (m), 1218 (vs), 1158 (w), 1041 (w), 754 (m).

*General procedure to obtain compounds***4a**–**4l**, The corresponding acid anhydride or acid chloride (1.5 equiv), DMAP (1.2 equiv), and Et_3_N (1.5 equiv) were added to a stirred solution of **1** (1 equiv) in CH_2_Cl_2_ (25 ml) at rt. After three hours, saturated aqueous NH_4_Cl/H_2_O was added, and the aqueous phase was extracted with CH_2_Cl_2_ (3 × 25 mL) before drying (Na_2_SO_4_) and removal of the solvent under reduced pressure. Purification by column chromatography (SiO2, 20:4 heptane/ethyl acetate) gave the desired products.

*(2-Methoxy-6,6-dimethyl-6H-benzo[c]chromen-9-yl)methyl acetate* (**4a**), Yield 92%. ^1^H and ^13^C shifts are shown in Table 2 and Table 3. HRMS-ESI+ (*m/z*): [M + H]^+^ calcd for C_19_H_21_O_4_, 313.1440; found, 313.1432. IR (cm^−1^): 2935 (vw, CH), 1739 (m, C=O), 1504 (m), 1425 (w), 1221 (vs), 1039 (w), 819 (vw).

*(2-Methoxy-6,6-dimethyl-6H-benzo[c]chromen-9-yl)methyl isobutyrate* (**4b**), Yield 67%. ^1^H and ^13^C shifts are shown in Table 2 and Table 3. HRMS-ESI+ (*m/z*): [M + H]^+^ calcd for C_21_H_25_O_4_, 341.1753; found, 341.1749. IR (cm^−1^): 2974 (w, CH), 1733 (s, C=O), 1504 (m), 1425 (w), 1218 (s), 1156 (s), 1041 (w), 821 (w), 737 (w).

*(2-Methoxy-6,6-dimethyl-6H-benzo[c]chromen-9-yl)methyl pivalate* (**4c**), Yield 62% ^1^H and ^13^C shifts are shown in Table 2 and Table 3. HRMS-ESI+ (*m/z*): [M + NH_4_]^+^ calcd for C_22_H_30_O_4_N, 372.2175; found, 372.2170. IR (cm^−1^): 2975 (m, CH), 1730 (s, C=O), 1505 (m), 1425 (w), 1282 (m), 1219 (m), 1155 (s), 1040 (w), 819 (vw).

*(2-Methoxy-6,6-dimethyl-6H-benzo[c]chromen-9-yl)methyl butyrate* (**4d**), Yield 58%. ^1^H and ^13^C shifts are shown in Table 2 and Table 3. HRMS-ESI+ (*m/z*): [M + NH_4_]^+^ calcd for C_21_H_28_O_4_N, 358.2018; found, 358.2014. IR (cm^−1^): 2966 (w, CH), 1735 (s, C=O), 1505 (m), 1425 (w), 1219 (s), 1173 (s), 1041 (w), 819 (vw), 738 (vw).

*(2-Methoxy-6,6-dimethyl-6H-benzo[c]chromen-9-yl)methyl 3-methylbutanoate* (**4e**), Yield 79%. ^1^H and ^13^C shifts are shown in Table 2 and Table 3. HRMS-ESI+ (*m/z*): [M + NH_4_]^+^ calcd for C_22_H_30_O_4_N, 372.2175; found, 372.2180. IR (cm^−1^): 2961 (m, CH), 1735 (vs, C=O), 1505 (m), 1425 (w), 1283 (w), 1218 (s), 1159 (m), 1094 (w), 1041 (w), 819 (w), 738 m).

*(2-Methoxy-6,6-dimethyl-6H-benzo[c]chromen-9-yl)methyl 3,3-dimethylbutanoate* (**4f**), Yield 92%. ^1^H and ^13^C shifts are shown in Table 2 and Table 3. HRMS-ESI+ (*m/z*): [M + NH_4_]^+^ calcd for C_23_H_32_O_4_N, 386.2331; found, 386.2326. IR (cm^−1^): 2958 (m, CH), 1733 (s, C=O), 1505 (m), 1425 (w), 1221 (vs), 1130 (s), 1042 (w), 819 (vw).

*(2-Methoxy-6,6-dimethyl-6H-benzo[c]chromen-9-yl)methyl hexanoate* (**4g**), Yield 78%. ^1^H and ^13^C shifts are shown in Table 2 and Table 3. HRMS-ESI+ (*m/z*): [M + NH_4_]^+^ calcd for C_23_H_32_O_4_N, 386.2331; found, 386.2325. IR (cm^−1^): 2957 (m, CH), 1736 (vs, C=O), 1505 (s), 1425 (m), 1219 (vs), 1159 (s), 1041 (w), 819 (vw).

*(2-Methoxy-6,6-dimethyl-6H-benzo[c]chromen-9-yl)methyl 3-cyclopentylpropanoate* (**4h**), Yield 67%. ^1^H and ^13^C shifts are shown in Table 2 and Table 3. HRMS-ESI+ (*m/z*): [M + NH_4_]^+^ calcd for C_25_H_34_O_4_N, 412.2488; found, 412.2487. IR (cm^−1^): 2948 (m, CH), 1735 (vs, C=O), 1504 (m), 1425 (m), 1219 (vs), 1158 (s), 1042 (w), 818 (vw).

*(2-Methoxy-6,6-dimethyl-6H-benzo[c]chromen-9-yl)methyl cyclohexanecarboxylate* (**4i**), Yield 57%. ^1^H and ^13^C shifts are shown in Table 2 and Table 3. HRMS-ESI+ (*m/z*): [M + NH_4_]^+^ calcd for C_24_H_32_O_4_N, 398.2331; found, 398.2330. IR (cm^−1^): 2934 (s, CH), 1732 (s, C=O), 1504 (m), 1425 (m), 1219 (vs), 1164 (s), 1131 (m), 1040 (m), 819 (w).

*(2-Methoxy-6,6-dimethyl-6H-benzo[c]chromen-9-yl)methyl acrylate* (**4j**), Yield 25%. ^1^H and ^13^C shifts are shown in Table 2 and Table 3. HRMS-ESI+ (*m/z*): [M + NH_4_]^+^ calcd for C_20_H_24_O_4_N, 342.1705; found, 342.1703. IR (cm^−1^): 2939 (w, CH), 1724 (vs, C=O), 1505 (s), 1425 (m), 1269 (s), 1218 (vs), 1177 (vs), 1114 (w), 1041 (m), 822 (vw).

*(2-Methoxy-6,6-dimethyl-6H-benzo[c]chromen-9-yl)methyl furan-2-carboxylate* (**4k**), Yield 96%. ^1^H and ^13^C shifts are shown in Table 2 and Table 3. HRMS-ESI+ (*m/z*): [M + NH_4_]^+^ calcd for C_22_H_24_O_5_N, 382.1654; found, 382.1653. IR (cm^−1^): 2979 (vw, CH), 1720 (s, C=O), 1572 (vw), 1504 (s), 1425 (m), 1294 (vs), 1218 (s), 1177 (s), 1113 (vs), 1041 (w), 947 (vw), 821 (w), 763 (m), 737 (m).

*(2-Methoxy-6,6-dimethyl-6H-benzo[c]chromen-9-yl)methyl benzoate* (**4l**), Yield 80%. ^1^H and ^13^C shifts are shown in Table 2 and Table 3. HRMS-ESI+ (*m/z*): [M + NH_4_]^+^ calcd for C_24_H_26_O_4_N, 392.1862; found, 392.1855. IR (cm^−1^): 2979 (vw, CH), 1718 (s, C=O), 1504 (m), 1425 (m), 1269 (vs), 1218 (m), 1109 (m), 1040 (w), 821 (vw), 712 (s).

*2-Methoxy-6,6-dimethyl-6H-benzo[c]chromene-9-carbaldehyde* (**5a**), Dess Martin periodinane 15% (1.4 mL, 0.7 mmol) was added to a stirred solution of **1** (60 mg, 0.2 mmol) in CH_2_Cl_2_ (50 mL) at rt. After five hours, saturated aqueous Na_2_S_2_O_3_/H_2_O was added, and the aqueous phase was extracted with CH_2_Cl_2_ (3 × 50 mL) before drying (Na_2_SO_4_) and removal of solvent under reduced pressure. Purification by column chromatography (SiO_2_, 20:1 CH_2_Cl_2_/methanol) gave (55 mg, 93%) as a yellowish wax. ^1^H and ^13^C shifts are shown in Table 2 and Table 3. HRMS-ESI+ (*m/z*): [M + H]^+^ calcd for C_17_H_17_O_3_, 269.1178; found, 269.1176. IR (cm^−1^): 2961 (vw, CH), 1699 (vs, C=O), 1500 (vs), 1426 (m), 1279 (w), 1219 (vs), 1154 (m), 1092 (w), 1039 (m), 946 (vw), 821 (m), 703 (vw).

*1-(2-Methoxy-6,6-dimethyl-6H-benzo[c]chromen-9-yl)ethan-1-ol* (intermediate in the synthesis of **5b**), MeMgI (3 M, 0.42 mL, 1.2 mmol) was added to **5a** (100 mg, 0.42 mg) in dry ethyl ether (5 mL), at 0 °C. After 20 h, saturated aqueous NH_4_Cl/H_2_O was added, and the aqueous phase was extracted with diethyl ether (3 × 5 mL). The organic product was washed with brine once before drying (Na_2_SO_4_) and removal of solvent under reduced pressure. Purification by column chromatography (SiO_2_, 20:5 hept/ethyl acetate) gave 61 mg, 51% as a yellowish wax. ^1^H-NMR (400 MHz, CDCl_3_) δ 7.64 (d, *J* = 2.0 Hz, 1H), 7.25 (d, *J* = 8.1 Hz, 1H), 7.24 (d, *J* = 1.1 Hz, 1H), 7.17 (d, *J* = 8.1 Hz, 1H), 6.86 (d, *J* = 8.8 Hz, 1H), 6.78 (dd, *J* = 8.7, 2.9 Hz, 1H), 4.90 (q, *J* = 6.4 Hz, 1H), 3.81 (s, 3H), 2.27 (s, 1H), 1.58 (d, *J* = 1.8 Hz, 6H), 1.50 (d, *J* = 6.5 Hz, 3H). ^13^C-NMR (100 MHz, CDCl_3_) δ 154.51, 146.85, 145.38, 139.16, 128.85, 125.28, 123.50, 123.10, 119.41, 118.75, 115.30, 108.05, 77.32, 70.31, 55.93, 27.45, 25.29. HRMS-ESI+ (*m/z*): [M + Na]^+^ calcd for C_18_H_20_O_3_, 307.1310; found, 307.1306. IR (cm^−1^): 3392 (br, OH), 2969 (m, CH), 1503 (s), 1381 (m), 1282 (m), 1216 (vs), 1157 (m), 1039 (m), 868 (w).

*1-(2-Methoxy-6,6-dimethyl-6H-benzo[c]chromen-9-yl)ethan-1-one* (**5b**), Celite (250 mg) and PCC (69 mg, 0.31 mmol) were added to a stirred solution of 1-(2-methoxy-6,6-dimethyl-6H-benzo[c]chromen-9-yl)ethan-1-ol (prepared as described above, 61 mg, 0.21 mmol) in dry CH_2_Cl_2_ (5 mL) under nitrogen, at rt. After 12 h diethyl ether (10 mL) was added and the mixture was filtered over a plug of silica gel and washed with ethyl acetate (50 mL) before removal of solvent under reduce pressure. Purification by column chromatography (SiO_2_, 20:4 hept/ethyl acetate) gave 51.5 mg, 87% as a yellowish wax. ^1^H and ^13^C shifts are shown in Table 2 and Table 3. HRMS-ESI+ (*m/z*): [M + H]^+^ calcd for C_18_H_19_O_3_, 283.1334; found, 283.1334. IR (cm^−1^): 2980 (vw, CH), 1684 (s, C=O), 1500 (vs), 1421 (m), 1244(s), 1212 (s), 1040 (m), 756 (vs).

*2-Methoxy-6,6-dimethyl-6H-benzo[c]chromene-9-carboxylic acid* (**5c**), Jones reagent (1 M, 0.5 mL) was added to **1** (60 mg, 0.22 mmol) in acetone (25 mL), at rt. After three hours, isopropanol (25 mL) was added, the organic phase was extracted with saturated aqueous NaHCO_3_ (3 × 25 mL). HCl (5% v/v) was added to the collected aqueous phase until acid pH (2), then mixture was extracted with EtOAc (2 × 100 mL) before drying (Na_2_SO_4_) and removal of solvent under reduced pressure, the procedure gave (25.5 mg, 41%) as yellowish wax. ^1^H and ^13^C shifts can be found in Table 2 and Table 3. HRMS-ESI+ (*m/z*): [M + H]^+^ calcd for C_17_H_17_O_4_, 285.1127; found, 285.1132. IR (cm^−1^): 2928 (vw, CH), 1691 (vs, C=O), 1502 (s), 1429 (w), 1302 (m), 1255 (s), 1041 (w), 948 (w), 758 (m).

*Methyl 2-Methoxy-6,6-dimethyl-6H-benzo[c]chromene-9-carboxylate* (**5d**), TMSCl (6.7 uL, 0.01 mmol) was added to **5c** (15 mg, 0.052 mmol) in methanol (2 mL) at rt. After 12 h, methanol was removed under reduced pressure and diethyl ether was added to the dry product; the resulting mixture was washed with saturated aqueous NaHCO_3_/H_2_O before drying (Na_2_SO_4_) and removal of solvent under reduced pressure. Purification by column chromatography (SiO_2_, 20:5 heptane/ethyl) gave (15 mg, 96%) as a colorless wax. ^1^H and ^13^C shifts are shown in Table 2 and Table 3. HRMS-ESI+ (*m/z*): [M + H]^+^ calcd for C_18_H_19_O_4_, 299.1283; found, 299.1290. IR (cm^−1^): 2950 (vw, CH), 1722 (vs, C=O), 1501 (s), 1412 (w), 1299 (m), 1282 (m), 1251 (vs), 1213 (m), 1156 (m), 1089 (w), 947 (vw), 871 (vw), 771 (vw).

*2-Methoxy-6,6-dimethyl-6H-benzo[c]chromene-9-carboxamide* (**5e**), Hydroxylamine (74.3 mg, 1.07 mmol) and copper acetate (20 mg, 0.11 mmol) were added to a stirred solution of **5a** (257 mg, 1.07 mmol) in dioxane (5 mL) at 104 °C. After 48 h, saturated aqueous NH_4_Cl/H_2_O was added, and the aqueous phase was extracted with ethyl acetate (3 × 5 mL) before drying (Na_2_SO_4_) and removal of solvent under reduced pressure. Purification by column chromatography on silica gel (SiO_2_, 99:1 ethyl acetate/methanol) and later on cross-linked dextran polymer beads (Sephadex LH50, 1:1 CHCl_3_/methanol) gave 22 mg, 7% as a yellowish wax. ^1^H and ^13^C shifts are shown in Table 2 and Table 3. HRMS-ESI+ (*m/z*): [M + H]^+^ calcd for C_17_H_18_NO_3_, 284.1287; found, 284.1286. IR (cm^−1^): 2964 (vw, CH), 1699 (vs, C=O), 1502 (s), 1434 (m), 1283 (m), 1262 (m), 1215 (s), 1040 (w), 749 (w).

*N**-hydroxy-2-methoxy-6,6-dimethyl-6H-benzo[c]chromene-9-carboximidamide* (**6**), Hydroxylamonium chloride (120 mg, 1.7 mmol) was added to **5a** (83.4 mg, 0.35 mmol) in EtOH (25 mL), at 80 °C. After 12 h, concentrated HCl (0.12 mL, 1.4 mmol) and zinc dust (57.2 mg, 0.90 mmol) were added slowly, at room temperature. After 30 min ammonia (0.5 mL) and 6 M NaOH (7 mL) where added until basic pH (10), the aqueous phase was extracted with CH_2_Cl_2_ (3 × 25 mL) before drying (Na_2_SO_4_) and solvent removal under reduced pressure. Purification by column chromatography (SiO_2_, 20:1 heptane/ethyl acetate) gave 5.5 mg, 6% as a yellowish wax. ^1^H and ^13^C shifts are shown in Table 2 and Table 3. HRMS-ESI+ (*m/z*): [M + H]^+^ calcd for C_17_H_19_N_2_O_3_, 299.1396; found, 299.1399. IR (cm^−1^): 2967 (m CH), 1682 (m, C=N), 1500 (s), 1214 (s), 1040 (m), 756 (vs).

### 3.3. Biological Assays

Evaluations against *Leishmania* parasites: Promastigotes of Leishmania-Leishmania: *L. amazonensis*, Clone 1, NHOM-BR-76-LTB-012 (Lma, donated by the Paul Sabatier Université, France) and Leishmania-Viannia: *L. braziliensis* M2904 C192 RJA (M2904, donated by Dr. Jorge Arévalo from Universidad Peruana Cayetano Heredia, Peru) [26]. All strains were cultured in Schneider’s insect medium, (pH 6.2) supplemented with 10% FBS and incubated at 26 °C. Medium changes were made every 72 h to maintain a viable parasitic population. Leishmanicidal activity was determined according to Williams with some modifications [27]. Samples were dissolved in DMSO (maximum final concentration 1%) at 10 mg/mL. Promastigotes in logarithmic phase of growth, at the concentration 3 × 10^6^ parasites/mL, were distributed (100 μL/well) in 96-well flat bottom microtiter plates. Samples with different concentrations (3.1–100 μg/mL) were added (100 μL). Miltefosine (3.1–100 μg/mL), was used as control drug [28]. Assays were performed in triplicates. The microwell plates were incubated for 72 h at 26 °C. After incubation, a solution of XTT (1 mg/mL) in PBS (pH 7.0 at 37 °C) with PMS (0.06 mg/mL) was added (50 μL/well), and incubated for 3 h at 26 °C. The optical density of each well was measured and the IC_50_ values calculated.

Evaluations against *Trypanosoma cruzi*: Cultures of *Trypanosoma cruzi* (epimastigotes, donated by the Parasitology Department of INLASA, Tc-INLASA), were maintained in medium LIT (pH 7.2), supplemented with 10% FBS and incubated at 26 °C. Medium changes were made every 72 h to maintain a viable parasitic population. Trypanocidal activity was determined according to Muelas-Serrano with some modifications [29]. Samples were dissolved in DMSO (maximum final concentration 1%) at 10 mg/mL. Epimastigotes in logarithmic phase of growth, at a concentration of 3×106 parasites/mL, were distributed (100 μL/well) in 96-well flat bottom microtiter plates. Samples at different concentrations (3.1–100 μg/mL) were added (100 μL). Benznidazol (3.1–100 μg/mL) was used as the control drug. Assays were performed in triplicates. The microwell plates were incubated for 72 h at 26 °C. After incubation, a solution of XTT (1 mg/mL) in PBS (pH 7.0 at 37 °C) with PMS (0.06 mg/mL) was added (50 μL/well) and incubated for 4 h at 26 °C. The optical density of each well was measured and the IC_50_ values were calculated.

Evaluations against RAW cells: The Raw 264.7 murine macrophage cell line was purchased from the American Type Culture Collection (ATCC-TIB71). The cells were maintained in DMEM-HG medium supplemented with 10% fetal bovine serum, 100 U/mL of penicillin and 100 μg/mL of streptomycin, and sodium bicarbonate (2.2 g/L) in humidified atmosphere at 37 °C with 5% CO_2_. Samples were prepared as described above and added (in 100 μL DMSO) at different concentrations (6.2–200 μg/mL). Medium blank, control drugs and cell growth controls were included to evaluate cell viability. The plates were incubated for 72 h at 37 °C with 5% CO_2_. After incubation for the indicated time, the cells were washed, after which 10 μL of Resazurin reagent (2.0 mM) was added. They were further incubated at 37 °C for 3 h in a humidified incubator. The IC_50_ values were assessed using a fluorometric reader (540 nm excitation, 590 nm emission) and the Gen5 software. All assays were performed in triplicate.

## 4. Conclusions

A more efficient synthetic protocol for preparing the antiparasitic plant metabolites pulchrol (**1**) and pulchral (**5a**) was developed. 25 analogues of **1** and **5a** with chemical variations of the benzylic alcohol functionality were prepared starting from **1**, and their antiparasitic activity against *T. cruzi*, *L. brasiliensis* and *L. amazoniensis* was tested. Although it was not possible to establish structure–activity relationships for such complex biological activities with the limited number of analogues available here, focusing on only one part of the core structure investigated, only some general suggestions can be made. In addition, nothing is known about the molecular targets that pulchrol (**1**) and its analogues interact with, it could be several and they may differ in the different organisms. In addition, changes in a chemical structure will give a new analogue somewhat different chemical properties. Changes in for example the solubility will affect the ability of a compound to penetrate cell membranes and thereby its capability to reach critical molecular targets in the required concentrations, and it is difficult to separate changes in the chemical properties with ability to target proteins. Hence, the better activity of the esters may be due to their higher lipophilicity, resulting in higher bioavailability, acting as ‘prodrugs’. In addition, we have no knowledge about the metabolic capacities of these organisms, which may affect the fate of the assayed compounds strongly. With that said, indications suggest that the benzylic oxygen in pulchrol is important for the anti-parasitic activity against *T. cruzi*, *L. amazoniensis* and *L. brasiliensis*, in some kind of hydrogen bond accepting capacity. The hydrogen bond donor role for pulchrol has been discarded for all cases, but hydrogen bond acceptor capability appears to be important from comparisons with the corresponding ethers, amines and esters analogues. In general, the more branched ethers and esters are more potent, suggesting a lipophilic pocket at the binding site. This presumed active site on which pulchrol and its analogues may be acting, is probably somewhat different in the different organisms. In *L. brasiliensis* there appear to be limited for branched chains of around five carbons, while *T. cruzi* and *L. amazoniensis* seem to have some flat hydrophobic regions that enhance the activity of aromatic and planar substituents. The selectivity as the SI is below or around 1 for most of the compounds, but is considerably higher in some examples. This is an important property to learn to understand, if compounds developed from this model ever should move forward into clinic trials. The best compound developed in this investigation is the vinyl ester **4j**. It shows potencies well below those of the positive controls and selectivity indexes of more than 5 for all three organisms. Another name for **4j** is an acrylic acid ester, and it is known that the acrylic acid derivatives may be weakly reactive as Michael acceptors. However, this is only relevant when an acrylic acid ester is bound to a pocket of a protein and presented to a highly reactive nucleophile (e.g., a thiol group of a cysteine, acting as an irreversible ‘covalent inhibitor’). If this is the explanation for the potency and selectivity of **4j** in this investigation, this type of analogues can eventually be used for the fishing out of the molecular targets of the parasites, and enable studies of such.

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
