# Peer review of "SAR:s for the Antiparasitic Plant Metabolite Pulchrol. 1. The Benzyl Alcohol Functionality"

_molecules, 2020, doi:10.3390/molecules25133058_

Round 1

Reviewer 1 Report

The article by Sterner et al may be of interest to the readers in tropical disease areas. The research is carried out well. This reviewer suggests few minor changes.

  1. Section 2.1, since authors have improved the synthetic route, and also for readers reference, a synthetic scheme could be added.
  2. Sentence 516-517, "Electrostatic interactions were indicated, involving for example the carbonyl group of the ester analogues, as well as a possible π-π interaction" (where & with what?) maybe a long shot. Authors may consider deleting the sentence.
  3. Authors may add, something like (a) after sentence 507-510, Hence, the better activity may be due to the effect of the esters resulting in higher bioavailability like in ester 'prodrugs'. (b) in line 529, (e.g. a thiol group of a cysteine, as an irreversible 'covalent inhibitor').

Author Response

  1. A synthetic Scheme describing the proparation of products has been added to section 2.1. Figure 1 has been changed accordingly.

2. This sentence has been deleted.

3. Corrections as suggested by the reviewer have been added.

Reviewer 2 Report

Dear Cecilia Li
Assistant Editor, MDPI

I am sending my review comments to the manuscript Molecules – 85 1900 - entitled SAR:s for the antiparasitic plant metabolite pulchrol. 1. The benzyl alcohol functionality.

Comments to the Authors

The manuscript prepared by Paola Terrazas and co-workes presents – studies reported about synthetic protocol for preparing the antiparasitic plant metabolites pulchrol (1) and pulchral (5a) was developed. 25 analogues of 1 and 5a with chemical variations of the benzylic alcohol functionality were prepared starting from 1, and their antiparasitic activity against T. cruzi, L. brasiliensis and L. amazoniensis was tested. 

  1. In part 2 Result and Discussion verse 63 the authors should comlete examples of literature

“Several synthetic strategies to prepare benzo[c]chromenes have been reported in the literaturÄ™[…..] the literaturÄ™ is given in werse 50 [18,19]. In my opinion this infprmation should be in verse 63.

2.       In the synthetic part, verse 241, I suggest giving full names of chemical compouds, insteat of scrutiny

Egxample 1. BH3-THF - Borane-tetrahydrofuran complex

The work presents research on a very important.However, it requires a slight correction, which I mentioned earlier ( comments to the Authors) 

After careful reading of this paper I am of the opinion that the work is suitable to Molecules.

Author Response

  1. The references have been moved to the suggested place.

2. Full names for the chemical compounds have been added to the synthetic part.